# Detailed Pathophysiology of Minimal Change Disease: Insights into Podocyte Dysfunction, Immune Dysregulation, and Genetic Susceptibility

**DOI:** 10.3390/ijms252212174

**Published:** 2024-11-13

**Authors:** Maja Roman, Michał Nowicki

**Affiliations:** Department of Nephrology, Hypertension, Transplantation and Internal Medicine, Central University Hospital, Medical University of Lodz, Pomorska 251, 92-213 Lodz, Poland; maja.roman@stud.umed.lodz.pl

**Keywords:** glomerular diseases, minimal change nephropathy, nephrotic syndrome, nephrin, podocin, biomarker

## Abstract

Minimal Change Disease (MCD) is a predominant cause of idiopathic nephrotic syndrome in the pediatric population, yet presents significant clinical challenges due to its frequent relapses and steroid resistance. Despite its relatively benign histological appearance, MCD is characterized by severe proteinuria, hypoalbuminemia, and edema, which may affect patient outcomes. Current treatment strategies primarily rely on corticosteroids, which are effective in inducing remission but are associated with high relapse rates, steroid resistance, and numerous long-term side effects, underscoring the need for more targeted and effective therapeutic approaches. This narrative review synthesizes current knowledge on the pathophysiological mechanisms underlying MCD, focusing on the following three critical areas: podocyte dysfunction, immune dysregulation, and genetic susceptibility. Podocyte dysfunction, particularly involving alterations in nephrin, plays a central role in the breakdown of the glomerular filtration barrier, leading to the characteristic proteinuria observed in MCD. Immune dysregulation, including the presence of autoantibodies against nephrin and other podocyte components, exacerbates podocyte injury and contributes to disease progression, suggesting an autoimmune component to the disease. Genetic factors, particularly mutations in the NPHS1 and NPHS2 genes, have been identified as significant contributors to disease susceptibility, influencing the variability in treatment response and overall disease severity. Understanding these mechanisms is crucial for developing targeted therapies that address the underlying causes of MCD rather than merely managing its symptoms. This review highlights the need for further research into these pathophysiological processes to pave the way for more personalized and effective treatment strategies, ultimately improving patient outcomes and reducing reliance on corticosteroids.

## 1. Introduction

### 1.1. Clinical Overview

Minimal Change Disease (MCD) is the most prevalent cause of nephrotic syndrome in children, accounting for approximately 70–90% of cases in this demographic [1,2,3,4]. In adults, MCD is less common but remains clinically significant, contributing to about 10–25% of nephrotic syndrome cases [5,6]. The worldwide annual incidence of MCD in children varies significantly by ethnicity and region: 2 per 100,000 in White children, 9.2 per 100,000 in Arabian children, and between 6.2 and 15.6 per 100,000 in Asian children. These figures emphasize the notable geographic and demographic differences in MCD prevalence. In adults, MCD is significantly rarer, though detailed incidence data for this group are limited [7]. Clinically, MCD is characterized by a sudden onset of nephrotic syndrome, including massive proteinuria, hypoalbuminemia, edema, and hyperlipidemia [3]. Despite its name, which suggests minimal pathological findings, MCD presents significant clinical challenges due to subtle histopathological changes that are often undetectable under light microscopy but evident through electron microscopy, where podocyte foot process effacement is the hallmark feature [8,9].

Current treatment strategies for MCD primarily rely on corticosteroids, which are effective in inducing remission in the majority of pediatric cases. However, this treatment is not without substantial limitations. A significant challenge in managing MCD is the high relapse rate, with many patients experiencing recurrent episodes requiring repeated courses of corticosteroids or additional immunosuppressive therapies. Furthermore, a subset of patients develops steroid resistance or dependence, leading to prolonged exposure to high-dose steroids. This long-term steroid use is associated with numerous adverse effects, including growth retardation, osteoporosis, obesity, hypertension, and increased susceptibility to infections, particularly in children [3,10,11,12,13,14].

In adults, MCD often presents with more severe symptoms and complications, such as acute kidney injury (AKI) and infections, and with higher risk renal function loss than in children. Treatment responses in adults are slower, with a high relapse rate leading to prolonged corticosteroid use, which carries significant risks, including osteoporosis, hypertension, and cardiovascular events. The response to corticosteroids in adults is typically slower and less predictable than in children, with some adults requiring several months to achieve remission. Furthermore, adults experience a higher relapse rate, and relapses are more likely to be steroid-resistant or dependent, requiring additional immunosuppressive therapies such as calcineurin inhibitors (e.g., cyclosporine or tacrolimus), cyclophosphamide, or rituximab to maintain disease control. Due to these challenges, adults with MCD are at a significantly higher risk of long-term complications, including CKD and progression to ESRD [15,16]. On the other hand, MCD is less common in adults, accounting for 10–25% of nephrotic syndrome cases [11,17,18].

In addition to differences in treatment response, the underlying pathophysiology may also vary between children and adults. For instance, while both populations share the characteristic effacement of podocyte foot processes observed on electron microscopy, the immune and molecular mechanisms contributing to disease progression and recurrence may differ. In adults, there is evidence to suggest that immune dysregulation, as seen with the overexpression of biomarkers like CD80 (B7-1), plays a more prominent role, potentially contributing to the greater likelihood of steroid resistance. Moreover, circulating permeability factors like soluble urokinase plasminogen activator receptor (suPAR) have been implicated in steroid-resistant cases of MCD in adults, further complicating disease management [19]. Prognostically, children generally fare better than adults. Pediatric patients with MCD tend to have fewer long-term complications and respond more consistently to standard steroid treatment. In contrast, adults are more prone to developing steroid dependence or resistance and are at greater risk of disease progression, particularly when complicated by comorbidities such as hypertension or secondary focal segmental glomerulosclerosis (FSGS). Consequently, while pediatric cases often resolve with minimal long-term consequences, adult cases require more aggressive treatment strategies and careful monitoring to prevent irreversible kidney damage [19].

These differences underscore the importance of tailored management strategies that account for age-specific risks and treatment responses. In children, the focus remains on managing relapses and minimizing steroid side effects, whereas, in adults, clinicians must be vigilant about monitoring for steroid resistance, systemic complications, and progression to CKD. Given these challenges, there is an urgent need for alternative therapeutic strategies that target the underlying pathophysiological mechanisms of MCD. Recent advances have highlighted the critical roles of podocyte injury, immune dysregulation, and genetic susceptibility in the disease’s progression [11,12]. Understanding these mechanisms is crucial for developing more effective and targeted treatments that could improve patient outcomes and reduce reliance on corticosteroids [9,10]. These differences between MCD in adults and children are compared in Table 1.

### 1.2. Pathophysiology of MCD

#### 1.2.1. Podocyte Dysfunction

Role of Podocytes in MCD:

Podocytes are specialized epithelial cells crucial to the glomerular filtration barrier in the kidneys, with their foot processes connected by the slit diaphragm allowing for selective blood filtration and the prevention of protein loss in the urine [22,23].

#### 1.2.2. Nephrin and the Slit Diaphragm

Nephrin, a crucial transmembrane protein, plays a pivotal role in the structural and functional integrity of the slit diaphragm. Alterations in nephrin, such as mislocalization, downregulation, or complete loss, are central to the pathophysiology of MCD [24,25,26,27,28,29]. Therapeutic strategies targeting nephrin stabilization and slit diaphragm preservation are under investigation to reduce proteinuria and improve clinical outcomes [30].

These changes compromise the slit diaphragm’s ability to act as a selective filtration barrier, leading to increased permeability and subsequent proteinuria. The loss of nephrin from the slit diaphragm correlates strongly with the severity of proteinuria observed in MCD patients. Furthermore, other proteins such as podocin, CD2AP, and NEPH1, which also contribute to slit diaphragm stability, can exacerbate disease progression when dysfunctional [31]. This understanding of nephrin and related proteins as key players in MCD pathogenesis not only highlights their role in disease but also suggests potential therapeutic targets aimed at preserving podocyte function and the integrity of the glomerular filtration barrier. As shown in Figure 1, genetic mutations, cytokine activity, and antibody formation lead to podocyte apoptosis and cytoskeletal damage, contributing to MCD pathophysiology.

#### 1.2.3. Clinical Implications

The central role of podocyte dysfunction, particularly through alterations in nephrin and the slit diaphragm, underscores the importance of targeted therapies that address these specific mechanisms, such as IL-13, TNF-alpha, IL-4, and the IL-33/ST2 axis, which are implicated in disrupting the glomerular filtration barrier [32]. Traditional treatments like corticosteroids, while effective in inducing remission, do not directly target the underlying podocyte dysfunction, often leading to high relapse rates and significant side effects. By contrast, therapies that stabilize nephrin or enhance its function could provide more sustained and effective treatment outcomes, reducing proteinuria and improving patient quality of life. Moreover, personalized medicine approaches that consider genetic mutations affecting nephrin could further refine treatment strategies, offering more individualized care based on a patient’s unique genetic profile [23,24,25,26,27,28,29,30,31,33].

### 1.3. Immune Dysregulation

#### 1.3.1. Immune Mechanisms in MCD

Immune dysregulation is another critical component of MCD pathogenesis. T-cell abnormalities and the production of autoantibodies against nephrin are central to this immune-mediated damage [34,35]. These autoantibodies can bind directly to nephrin, leading to its dysfunction or loss from the slit diaphragm, which exacerbates the breakdown of the filtration barrier and contributes to the development of nephrotic syndrome. The role of T-cell-secreted permeability factors, though not fully understood, further highlights the complexity of immune involvement in MCD, suggesting that the disease may have an autoimmune component [36].

#### 1.3.2. Cytokine Involvement

Cytokines, particularly IL-13, play a significant role in the immune-mediated damage observed in MCD. IL-13 affects podocyte gene expression, leading to structural and functional changes that compromise the integrity of the glomerular filtration barrier [37,38]. Additionally, IL-13 influences the differentiation of T-helper 2 (Th2) cells, which promote B-cell activation and autoantibody production, creating a pro-inflammatory environment detrimental to podocyte health. The involvement of other cytokines, such as TNF-alpha and IL-4, further contributes to this immune dysregulation, underscoring the need for therapies that target these specific pathways [39,40]. The study highlights the significance of the IL-33/ST2 axis in the pathophysiology of MCD. IL-33, a cytokine involved in type 2 immune responses, is found to be highly expressed in the podocytes of MCD patients, suggesting its role in disrupting the glomerular filtration barrier and contributing to proteinuria. Additionally, elevated levels of soluble ST2 (sST2), the decoy receptor for IL-33, correlate with disease severity, indicating the active involvement of this axis in MCD progression. In the context of MCD, IL-33 may be secreted by podocytes in response to stress or injury, leading to the activation of ST2-positive immune cells, such as type 2 innate lymphoid cells (ILC2s) and Th2 cells. These cells can produce cytokines like IL-13, which have been implicated in the exacerbation of podocyte injury and proteinuria. Immunomodulatory therapies for MCD aim to inhibit specific cytokines like IL-13, IL-33, and TNF-alpha, or immune cells such as ILC2s and Th2 cells, to reduce inflammation and podocyte injury. These targeted approaches help preserve kidney function by disrupting the pro-inflammatory pathways driving MCD progression [32].

#### 1.3.3. Clinical Relevance

The insights into immune mechanisms and cytokine involvement in MCD have significant implications for treatment. Current therapies, like corticosteroids, broadly suppress the immune response but are associated with considerable side effects and high relapse rates. Targeted immunomodulatory therapies that inhibit specific cytokines or prevent autoantibody production offer a promising alternative. For example, therapies targeting IL- 13 or its receptor could reduce podocyte injury and provide a more refined and effective treatment approach, particularly for patients who are steroid-resistant or experience frequent relapses [32,34,35].

### 1.4. Genetic Susceptibility

#### 1.4.1. Genetic Factors in MCD

Genetic susceptibility plays a pivotal role in MCD, particularly in determining an individual’s risk of developing the disease and their response to treatment. Mutations in key genes such as NPHS1 and NPHS2, which encode the proteins nephrin and podocin, respectively, are well-documented contributors to MCD pathogenesis A [33,38,39]. These mutations disrupt the slit diaphragm’s structural integrity, leading to severe proteinuria and influencing the severity and treatment response of the disease. Other genes, such as WT1 and ACTN4, have also been implicated, further highlighting the complex genetic underpinnings of MCD [41,42,43,44].

#### 1.4.2. Impact of Genetic Variants

The presence of these genetic mutations often correlates with more severe disease phenotypes, early onset, and a higher likelihood of resistance to conventional treatments such as corticosteroids. For instance, patients with NPHS1 or NPHS2 mutations may experience frequent relapses and progression to end-stage renal disease (ESRD) if not managed effectively. Additionally, genetic polymorphisms in cytokine-related genes, such as IL-13 and TNF-alpha, may modulate the severity of immune activation, further complicating treatment [37,38,39,40].

#### 1.4.3. Implications for Personalized Medicine

The identification of genetic mutations associated with MCD has significant implications for personalized medicine. Genetic screening can help predict disease risk and guide treatment decisions, allowing for early and aggressive intervention in high-risk patients. Furthermore, therapies that target the specific molecular pathways affected by these mutations, such as gene therapy or targeted biologics, hold promise for more effective and individualized treatment strategies. Ultimately, incorporating genetic information into clinical practice could transform the current management of MCD, leading to better outcomes and a reduced reliance on corticosteroids [45,46].

### 1.5. Biomarkers in MCD

Biomarkers play an essential role in elucidating the pathophysiological mechanisms underlying MCD and in guiding its clinical management. Among the most pivotal biomarkers is nephrin, a transmembrane protein that is a crucial component of the slit diaphragm in podocytes. Nephrin’s structural integrity is vital for maintaining the selective permeability of the glomerular filtration barrier. In MCD, the redistribution or downregulation of nephrin has been extensively documented, and this alteration is strongly associated with the onset of proteinuria. The identification of autoantibodies against nephrin in some patients further suggests an autoimmune component to MCD, wherein these autoantibodies may directly contribute to podocyte injury by disrupting nephrin’s function or promoting its mislocalization within the podocyte [47]. Podocin, another critical protein in the slit diaphragm, works closely with nephrin to maintain the structure and function of the glomerular filtration barrier. Mutations or the altered expression of podocin can exacerbate the vulnerability of the glomerular barrier, leading to increased permeability and proteinuria. The involvement of podocin in MCD underscores the importance of podocyte health and integrity in this disease [48].

Additionally, immune-related biomarkers such as CD80 (B7-1) have been implicated in MCD. CD80 is typically expressed on the surface of activated immune cells, but its aberrant expression in podocytes has been linked to the disease’s pathogenesis. Elevated levels of CD80 in the urine or on podocyte surfaces during active disease phases suggest that immune dysregulation plays a crucial role in MCD. This biomarker not only aids in diagnosing MCD but also offers potential therapeutic targets, especially in cases where conventional treatments are insufficient [49,50]. Cytokines, particularly IL-13, have also been highlighted in recent studies as significant contributors to the disease process in MCD. IL-13 is known to influence the immune response and has been shown to induce podocyte injury by downregulating nephrin and other crucial slit diaphragm proteins. Elevated IL-13 levels may be indicative of an active immune-mediated process contributing to disease exacerbation, making it a potential marker for disease activity and a target for novel therapeutic interventions [51]. Furthermore, the identification of circulating permeability factors, such as suPAR, adds another layer of complexity to the understanding of MCD. These factors can influence podocyte function and contribute to the pathogenesis of proteinuria by inducing podocyte effacement and disrupting the actin cytoskeleton. The role of these circulating factors is particularly significant in cases where MCD does not respond well to steroid therapy, suggesting an alternative pathway for disease progression that might require different therapeutic approaches [50]. Recent advances have also identified various RNA molecules as emerging biomarkers in MCD. Long non-coding RNAs (lncRNAs), for example, are involved in regulating gene expression at multiple levels, including chromatin modification, transcription, and post-transcriptional processing. Dysregulation of specific lncRNAs in MCD can disrupt podocyte function and immune responses, contributing to disease progression by affecting the expression of key genes involved in maintaining the glomerular filtration barrier [52]. Similarly, circular RNAs (circRNAs), which are stable, covalently closed loop structures, have been implicated in the regulation of podocyte apoptosis and proliferation, both critical processes in the development of proteinuria in MCD [53]. Small interfering RNAs (siRNAs) [54] and transfer RNA-derived fragments (tRFs) [55] represent additional classes of RNA biomarkers under investigation. siRNAs can silence gene expression by degrading mRNA, thus preventing the production of specific proteins, and are being explored for their role in understanding gene dysregulation in MCD. tRFs, generated from precursor or mature tRNAs, have been linked to the regulation of gene expression and cellular stress responses. Though still under investigation, these RNA fragments may provide further insights into the molecular mechanisms underlying MCD. Pseudogene-derived RNAs [56,57], often considered non-functional remnants of genes, have shown potential as biomarkers in MCD. These RNAs can regulate other genes by acting as miRNA decoys or through other mechanisms, potentially influencing disease activity and treatment response. One of the RNA biomarkers extensively studied in MCD is microRNAs (miRNAs). These small non-coding RNAs regulate gene expression at the post-transcriptional level. In MCD, specific miRNAs, such as miR-193a, have been identified as being upregulated and involved in promoting podocyte injury by targeting and downregulating critical podocyte-specific proteins like nephrin and podocin [58]. This miRNA-mediated disruption of podocyte architecture contributes significantly to the proteinuria observed in MCD. Additionally, altered mRNA expression profiles related to key podocyte proteins and immune markers have been observed in MCD. For instance, studies have demonstrated changes in the mRNA levels encoding nephrin, podocin, and other slit diaphragm components, as well as elevated mRNA levels of immune-related genes like CD80 (B7-1) during active disease phases. These alterations reflect the immune dysregulation characteristic of MCD and offer potential targets for diagnostic and therapeutic interventions. Alpha-1-microglobulin (A1M) has been studied as a marker for tubular dysfunction, and its urinary excretion is often used to assess kidney function, particularly in glomerular diseases like MCD [59]. The significance of cytokines’ involvement in the pathogenesis of MCD can be found in Table 2. The diagnostic values of the biomarkers are found in Table 3.

#### 1.5.1. Current Treatment Challenges

The management of MCD heavily relies on corticosteroid therapy due to its effectiveness in inducing remission. However, the extensive use of corticosteroids is accompanied by several significant challenges. A substantial number of patients develop steroid resistance, particularly in conditions like Focal Segmental Glomerulosclerosis (FSGS), where corticosteroids fail to induce remission, leading to disease progression and poor long-term outcomes [25,64,65,66]. MCD and FSGS share overlapping clinical and histopathological features, particularly in cases of steroid-resistant nephrotic syndrome. Both conditions involve podocyte injury and proteinuria; however, FSGS is characterized by segmental scarring (sclerosis) of glomeruli, which is typically absent in MCD. MCD can progress to FSGS in certain patients, especially those who develop steroid resistance. In such cases, corticosteroids and other immunosuppressive therapies often fail to induce remission, leading to progressive renal function decline. The transition from MCD to FSGS is associated with worse prognoses, including an increased risk of CKD and ESRD, which emphasizes the need for targeted therapies beyond corticosteroids for patients with steroid-resistant forms of the disease [48]. Additionally, many patients experience steroid dependence, characterized by frequent relapses upon reducing or discontinuing the therapy, necessitating the prolonged use of corticosteroids. Long-term corticosteroid therapy is associated with severe side effects, including growth retardation in children, osteoporosis, hypertension, hyperglycemia, and an increased risk of infections. These adverse effects complicate the management of the disease and significantly impact patients’ quality of life, underscoring the urgent need for alternative therapeutic strategies [3,11,12].

#### 1.5.2. Potential for Targeted Therapies

Advancements in the understanding of the pathophysiology of MCD, particularly the role of podocyte injury and immune dysregulation, have driven the development of targeted therapies aimed at overcoming the limitations of corticosteroids. Among these, rituximab, an anti-CD20 monoclonal antibody, has shown significant promise in reducing relapses and inducing remission in steroid-resistant and steroid-dependent nephrotic syndrome [48,67,68]. Tacrolimus, another immunosuppressant, is effective in inducing remission in steroid-resistant cases but requires careful monitoring due to potential nephrotoxicity [69,70,71]. Mycophenolate mofetil is also being explored as a steroid-sparing agent, offering a viable option for maintaining remission with fewer side effects [72,73,74].

Additionally, Active Vitamin D Analogs, such as alfacalcidol, are being studied for their potential benefits in modulating the immune system and improving bone health, particularly when used in combination with reduced-dose steroids [75,76,77]. Furthermore, novel therapies like Ripertamab [78] and TRPC6 channel inhibitors [43,44] are being investigated for their ability to target specific molecular pathways involved in MCD. These emerging therapies, alongside ongoing research into gene therapy and other biologics, represent a significant shift towards more effective and safer treatment options that could potentially transform the therapeutic landscape for MCD patients. VB119 was another promising therapeutic candidate under investigation, targeting different aspects of the disease; however, its clinical trial was terminated, likely due to concerns about efficacy or safety [79,80]. This highlights the challenges in developing new therapies, but ongoing research continues to seek more effective and safer treatment options.

#### 1.5.3. Clinical Trials and Future Directions

The development of these targeted therapies is supported by a growing body of clinical research, with numerous trials underway to evaluate their safety and efficacy in MCD patients. These trials are crucial for determining the potential of these therapies to replace or supplement corticosteroids, reducing the burden of side effects and improving long-term outcomes. As these therapies move closer to clinical application, they have the potential to revolutionize the treatment of MCD, offering hope for more personalized and effective management strategies.

In summary, the emergence of targeted therapies for MCD represents a significant advancement in the treatment of this challenging disease. By focusing on the specific mechanisms underlying MCD, these therapies offer the potential to provide more effective and safer treatment options, reducing the reliance on corticosteroids and improving the quality of life for patients. The future of MCD treatment lies in these targeted approaches, which hold the promise of transforming the therapeutic landscape and achieving better outcomes for patients with MCD [81].

### 1.6. Future Directions in Precision Medicine

As these clinical trials progress, there is growing interest in applying precision medicine to MCD. The concept of precision medicine involves tailoring treatments based on individual patient characteristics, including genetic mutations, biomarker profiles, and disease phenotypes. Genetic testing and biomarker profiling could soon become integral to clinical practice, guiding the selection of therapies that are most likely to be effective for specific patient subgroups. For example, patients identified with mutations in the NPHS1 or NPHS2 genes might benefit from early intervention with targeted therapies such as rituximab or TRPC6 inhibitors, potentially preventing disease progression and minimizing the side effects associated with prolonged corticosteroid use. The integration of genetic and biomarker data into treatment decision making represents a significant step forward in the personalization of MCD management, offering the potential for improved outcomes and reduced treatment-related complications [25,64,65,82]. Additionally, podocin and uPAR have been identified as effective biomarkers in distinguishing FSGS from MCD in pediatric renal biopsies, with podocin showing a sensitivity of 73.3% and specificity of 86.7% [83]. A comparison of the therapies is found in Table 4.

### 1.7. Expanding the Therapeutic Arsenal

Beyond the therapies currently in clinical trials, ongoing research is likely to yield new therapeutic candidates in the coming years. Advances in biotechnology, such as analyzing the gene expression in preserved biopsy samples, have identified specific markers for MCD, such as IL7RA and IL12RB1, which could serve as new targets for treatment [85]. This research emphasizes the value of understanding the molecular details of MCD to develop precise and targeted therapies. For instance, CRISPR-Cas9 technology could one day be used to correct genetic mutations in podocyte-related genes, offering a potential cure for genetically predisposed forms of MCD [86,87].

Exploring novel immunomodulatory agents and therapies that influence microbiome composition or epigenetic factors presents promising strategies for MCD, addressing its multifactorial nature more effectively than traditional treatments [88,89].

### 1.8. Research Gaps and Future Studies

Despite significant advancements in understanding MCD, several key research gaps remain that must be addressed to further improve treatment outcomes. Identifying these gaps is crucial for guiding future research efforts and ensuring that they focus on the most critical aspects of the disease.

### 1.9. Gene–Environment Interactions

One of the most significant gaps in the current understanding of MCD involves the interactions between genetic predispositions and environmental factors. While there is substantial evidence linking certain genetic mutations, such as those in the NPHS1 and NPHS2 genes, to the development of MCD, the role of environmental triggers in modulating these genetic risks is less clear [25]. Future research should focus on identifying specific environmental factors, such as infections, toxins, or dietary components, that may interact with genetic mutations to trigger the onset of MCD or influence its progression [21]. Understanding these interactions could lead to the development of preventive strategies or interventions that mitigate the impact of environmental risks in genetically susceptible individuals. A list of genetic mutations and their implications can be found in Table 5.

### 1.10. Mechanisms of Steroid Resistance

Steroid resistance remains a major challenge in the management of MCD, with some patients failing to respond to conventional corticosteroid therapy. The mechanisms underlying steroid resistance are not fully understood, and this gap in knowledge limits the ability to develop effective alternative therapies [21]. Future studies should aim to elucidate the molecular and cellular pathways that contribute to steroid resistance, focusing on identifying biomarkers that can predict which patients are likely to be resistant to steroid therapy. Such biomarkers could guide the development of personalized treatment plans, allowing for the earlier implementation of alternative therapies in patients who are unlikely to respond to steroids [25,34,65,83,90].

### 1.11. Long-Term Efficacy and Safety of Emerging Therapies

While several new therapies are currently under investigation for the treatment of MCD, including biologics and small molecules that target specific pathways involved in the disease, there is a lack of long-term data on their efficacy and safety. It is essential to conduct longitudinal studies that track the outcomes of patients treated with these emerging therapies over extended periods. These studies should not only assess the ability of these therapies to sustain remission and prevent relapse but also monitor for potential long-term adverse effects that may not be apparent in shorter clinical trials. Understanding the long-term impact of these treatments will be critical for their successful integration into clinical practice [63,93,94].

**Figure 1 ijms-25-12174-f001:**
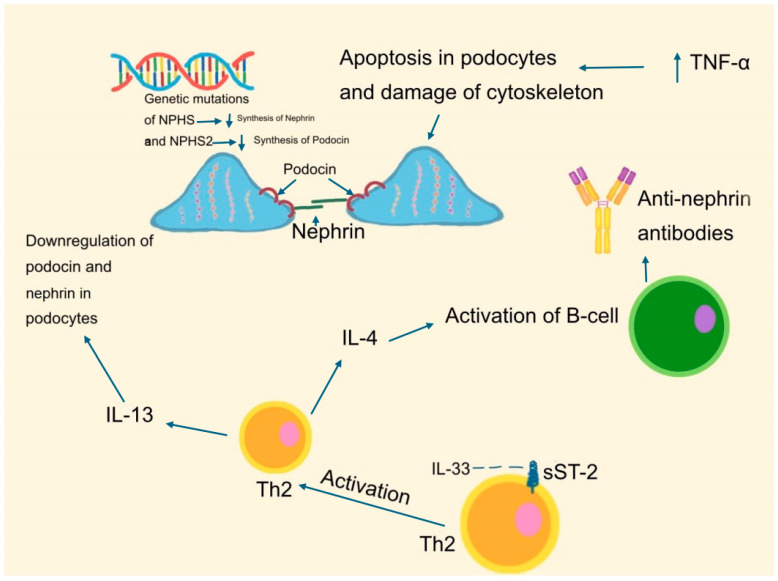
Pathophysiological mechanisms in Minimal Change Disease (MCD). Genetic mutations in NPHS1 and NPHS2 genes resulting in the synthesis of nephrin and podocin, essential for podocyte integrity. Inflammatory cytokines (e.g., IL-4 and IL-13) and anti-nephrin antibodies contributing to podocyte apoptosis and cytoskeletal damage, disrupting the glomerular filtration barrier. Activation of B-cells and Th2 immune response pathways further exacerbating podocyte injury and proteinuria [23,24,25,26,27,28,29,30,31,33,37,38,39,40,50,58,93,95].

### 1.12. Novel Therapeutic Targets

Another important area for future research is the identification of novel therapeutic targets that go beyond the currently explored pathways. While much of the current research focuses on podocyte injury and immune dysregulation, other molecular mechanisms may also play a role in the pathogenesis of MCD [8]. Exploring these less understood pathways could reveal new targets for therapy, potentially leading to the development of treatments that are more effective for patients who do not respond to existing options.

Additionally, research into gene therapy and other innovative technologies may offer new avenues for treating MCD at the molecular level, addressing the root causes of the disease [96,97].

### 1.13. Integration of Biomarkers into Clinical Practice

The development of reliable biomarkers for MCD is crucial for advancing personalized medicine in the management of the disease.

However, despite progress in identifying potential biomarkers, their integration into clinical practice remains limited. Future research should focus on validating these biomarkers in large, diverse patient populations and developing standardized protocols for their use in diagnosis, prognosis, and monitoring treatment response. Integrating biomarkers into routine clinical practice could significantly improve the precision of MCD management, leading to better outcomes for patients through more personalized and timely therapeutic interventions [96,97]. The comparison of the responses to therapeutic treatment depending on genetic factors and biomarkers can be found in Table 6.

### 1.14. Comparative Analysis of MCD in Global Renal Biopsy Studies

MCD is a common cause of nephrotic syndrome, particularly in children, but its prevalence varies significantly across regions and age groups. In Poland, studies reveal that MCD constitutes 5.5% of nephrotic syndrome cases in adults, while it accounts for a much larger proportion, around 22%, in pediatric cases. This trend is consistent with global findings, where MCD remains a leading cause of childhood nephrotic syndrome but declines in frequency among adults, making way for other glomerulopathies like membranous nephropathy (MN) and FSGS [100].

In Spain, MCD is similarly prevalent in children, constituting 39.5% of pediatric nephrotic syndrome cases, but declines significantly in adults, where MN dominates at 24.2% in the adult population and 28% in the elderly. The Spanish Glomerulonephritis Registry data underscore the shift from MCD to MN as patients age, reflecting the general global pattern where MCD, though significant in younger populations, becomes overshadowed by other glomerular diseases in older groups [101].

In northeast China, a 10-year retrospective study identified MCD in 12.7% of all renal biopsy cases, with a notable prevalence in adults aged 20 to 39, constituting 41.3% of the MCD cases. This is slightly higher than the figures from Spain and Poland for the adult population, possibly reflecting regional genetic or environmental factors influencing the pathology of kidney diseases. Interestingly, the study also noted an increasing trend of MN in adults, similar to the trend seen in other regions, which suggests a possible environmental influence, such as increased pollution exposure, contributing to the rise of MN as a dominant pathology [102].

In South India, however, MCD does not have the same prominence as seen in Europe or China. Here, IgA nephropathy (IgAN) is the leading cause of renal disease, particularly in younger populations, accounting for 24.4% of biopsies [103]. This is in stark contrast to the data from Spain, where IgAN is more frequently associated with asymptomatic urinary abnormalities rather than nephrotic syndrome. The lower frequency of MCD in India may be attributable to different environmental exposures or diagnostic criteria, as well as genetic factors prevalent in the Indian population.

### 1.15. Global Trends and Clinical Implications

While MCD is a dominant cause of nephrotic syndrome in children worldwide, the variability in its prevalence across adult populations emphasizes the importance of regional factors. For instance, in Poland, the relatively lower incidence of MCD in adults compared to China suggests that genetic and environmental factors may play significant roles. In Spain, the shift from MCD to MN as the primary cause of nephrotic syndrome in adults aligns with global trends seen in other high-income countries [101]. In northeast China, the rising incidence of MN and the sustained prevalence of MCD in younger adults indicate shifting epidemiological patterns, possibly linked to changes in environmental exposures, such as air pollution. Studies have highlighted that prolonged exposure to particulate matter (PM2.5) is associated with an increased risk of developing MN, a trend that might explain the rising MN cases in China. Meanwhile, in South India, the predominance of IgA nephropathy reflects a regional pattern observed across Asia, where IgAN is more common than in Western countries. The lower prevalence of MCD in South India compared to Europe and China underscores the diverse etiologies and pathophysiologies of nephrotic syndrome globally. These differences highlight the need for region-specific diagnostic and treatment approaches, as MCD’s response to steroid therapy may differ across populations based on the underlying genetic and environmental factors [101,102,103].

## 2. Conclusions

MCD remains a significant challenge for clinicians, particularly in pediatric populations, where it is the leading cause of nephrotic syndrome. Despite the effectiveness of corticosteroids in inducing remission, the high rates of relapse, steroid resistance, and the substantial side effects associated with long-term steroid use underscore the urgent need for more targeted and safer therapeutic approaches [3,10,11,12]. This narrative review has highlighted the critical role of podocyte dysfunction, immune dysregulation, and genetic susceptibility in the pathogenesis of MCD. The integration of these pathophysiological insights into clinical practice is crucial for developing more personalized and effective treatment strategies. Emerging therapies that target specific aspects of the disease, such as B-cell depletion, cytokine inhibition, and actin cytoskeleton stabilization, represent a promising shift away from the broad-spectrum immunosuppression currently employed and may address currently unmet medical needs in this disease [8,15,84]. However, significant research gaps remain, particularly in understanding the gene environment interactions that may trigger or exacerbate MCD, the mechanisms underlying steroid resistance, and the long-term efficacy and safety of new therapies. Addressing these gaps through continued research is essential for advancing our understanding of MCD and improving patient outcomes. The future of MCD management lies in the adoption of precision medicine approaches that leverage genetic, biomarker, and phenotypic information to tailor treatment to the individual patient. This personalized approach has the potential to not only enhance treatment efficacy but also to minimize the adverse effects associated with traditional therapies. As new therapies emerge from clinical trials, their integration into clinical practice will require careful consideration of their long-term impact on patient health and quality of life [3,10,63,93,94].

In conclusion, the insights gained from this review underscore the importance of a continued focus on developing and implementing targeted therapies that address the underlying causes of MCD. By moving towards a more personalized and mechanistically informed approach to treatment, we can hope to achieve better outcomes for patients and ultimately transform the therapeutic landscape for this challenging disease. A comparison of MCD and other nephrotic syndromes’ immune dysregulation is provided in Table 7.

## Figures and Tables

**Table 1 ijms-25-12174-t001:** Comparison of MCD in adults vs. children.

Aspect.	Adults [11,17,18,20]	Children [20,21]
Prevalence	Less common (10–25% of nephrotic syndrome cases)	More common (70–90% of nephrotic syndrome cases)
Clinical Presentation	More severe with higher incidence of acute kidney injury and infections; more likely to present with comorbidities	Typically presents with classic nephrotic syndrome (massive proteinuria, hypoalbuminemia, edema, and hyperlipidemia)
Response to Corticosteroids	Slower response; higher likelihood of steroid resistance or dependence	Generally rapid and favorable response to corticosteroids
Relapse Rate	High; frequent relapses common, often requiring prolonged or repeated steroid therapy	High relapse rate but often responsive to repeat steroid therapy
Complications	Higher risk of long-term complications from steroids (e.g., osteoporosis, hypertension, and cardiovascular issues)	Complications include growth retardation, obesity, and infections related to steroid use
Histopathology	Subtle changes often detectable only through electron microscopy (podocyte foot process effacement)	Similar histopathological findings with minimal changes under light microscopy
Prognosis	More complex and less predictable; risk of progression to chronic kidney disease in severe cases	Generally favorable with good long-term outcomes, although relapses are common
Additional Therapies	May require additional immunosuppressive therapies like calcineurin inhibitors or rituximab	Additional therapies less commonly needed but used in cases of frequent relapse or steroid resistance
Mortality and Morbidity	Higher due to complications and comorbidities	Lower overall, with most children achieving long-term remission

**Table 2 ijms-25-12174-t002:** Cytokines implicated in MCD and their potential clinical importance.

Cytokine	Role in MCD	Associated Clinical Outcomes	Potential Therapeutic Targets
IL-13	Alters podocyte gene expression, contributing to cytoskeletal changes and proteinuria [47].	Severe proteinuria; progression to nephrotic syndrome [47].	Anti-IL-13 monoclonal antibodies; cytokine pathway inhibitors [47].
TNF-alpha	Induces podocyte apoptosis and disrupts the glomerular basement membrane integrity [47,60].	Increased proteinuria, inflammation, and kidney injury [47,60].	TNF-alpha inhibitors (e.g., etanercept and infliximab) [47,60].
IL-4	Promotes Th2 cell differentiation, enhancing B-cell activation and autoantibody production [47,60].	Immune dysregulation; potential contribution to steroid resistance [47,60].	IL-4 receptor antagonists; Th2-targeted immunomodulatory therapies [47,60].
IL-6	Involved in the acute-phase response, contributing to inflammation and immune activation [47,60].	Chronic inflammation; worsening of nephrotic syndrome [47,60].	IL-6 inhibitors (e.g., tocilizumab); anti-inflammatory therapies [47,60].
TGF-beta	Promotes fibrosis and podocyte injury, contributing to chronic kidney disease progression [47].	Increased risk of fibrosis, chronic kidney damage, and progression to ESRD [47].	TGF-beta pathway inhibitors; anti-fibrotic agents [47].

End-stage renal disease [ESRD].

**Table 3 ijms-25-12174-t003:** Biomarkers for MCD diagnosis and monitoring.

Biomarker	Diagnostic Utility	Monitoring Utility	Prognostic Value
Urinary Nephrin	Early marker of podocyte injury helps in diagnosing MCD [61].	Monitoring disease activity correlates with the extent of proteinuria [61].	Higher levels may indicate ongoing podocyte damage and disease severity [61].
Serum Albumin	Indicative of nephrotic syndrome when levels are low [21].	Used to monitor treatment response, especially to corticosteroids [21].	Persistently low levels suggest poor prognosis and potential for relapse [21].
Podocalyxin	Marker of podocyte detachment, aiding in MCD diagnosis [62].	Reflects podocyte injury, used to assess treatment effectiveness [62].	Elevated levels may indicate active disease and risk of progression [62].
Urinary CD80	Associated with immune activation in MCD, and a potential diagnostic tool [51].	Monitors response to immunosuppressive therapy, especially in relapse [51].	Elevated levels correlate with immune dysregulation, indicating poor response to steroids [51].
Alpha-1-microglobulin	Reflects tubular protein reabsorption capacity, and useful in early diagnosis [59,63]	Monitors kidney function and the effectiveness of therapy [59,63].	Elevated levels may indicate tubular damage and worse long-term outcomes [59,63].

Minimal Change Disease [MCD].

**Table 4 ijms-25-12174-t004:** Comparison of corticosteroid therapy vs. emerging targeted therapies in MCD.

Therapy Type	Mechanism of Action	Efficacy in Inducing Remission	Side Effects	Long-Term Outcomes	Examples of Emerging Therapies
Corticosteroid Therapy	Broad immune suppression [21,84].	High in most patients [21].	Growth retardation, osteoporosis, obesity, hypertension, and increased infection risk [21,85].	High relapse rates, steroid resistance, and significant side effects over time [21,84,85].	Not applicable [21]
B-Cell-Targeted Therapies	Depletion of B-cells; reduction in pathogenic antibodies (e.g., rituximab) [21,84].	Effective in steroid-resistant/dependent patients [84].	Well-tolerated; infusion reactions possible [84].	Lower relapse rates compared to steroids; long-term safety data needed [84].	Rituximab [84]
Actin Cytoskeleton Stabilizers	Stabilizing podocyte structure to prevent foot process effacement [46].	Early-stage research; potential efficacy [46].	Few side effects expected based on mechanism of action [46].	Could improve podocyte function and reduce disease progression [46].	Small molecules targeting actin dynamics [46]
TRPC6 Inhibitors	Inhibition of TRPC6 to prevent calcium-induced podocyte injury [46].	Promising in preclinical and early trials [46].	Specific side effects depending on TRPC6 inhibition [46].	Potential to prevent podocyte injury and reduce proteinuria long-term [46].	TRPC6 inhibitors (in development) [46]

**Table 5 ijms-25-12174-t005:** Known genetic mutations associated with MCD and their clinical implications.

Gene	Mutation Type	Associated Protein	Clinical Implications	Potential Targeted Therapies
*NPHS1*	Missense, nonsense, and splice site mutations [33,42,90].	Nephrin [33,42,90,91]	Early-onset nephrotic syndrome, severe proteinuria, resistance to corticosteroids, and progression to ESRD [42,90,91].	Gene therapy, nephrin replacement strategies, early intervention, and nephrin stabilization [42,90,91].
*NPHS2*	Missense mutations, nonsense mutations, and deletions [42,90,91].	Podocin [42,90,91]	Autosomal recessive steroid-resistant nephrotic syndrome and increased risk of progression to ESRD [42,90,91].	Gene therapy, podocin stabilization, and precision medicine approaches [42,90,91].
*WT1*	Missense mutations, deletions, and point mutations [91].	Wilms tumor 1 protein [91]	Congenital nephrotic syndrome, steroid resistance, associated with Wilms tumor, and gonadal dysgenesis [91].	WT1-targeted therapies, personalized care, early genetic screening, and tumor monitoring [91].
*ACTN4*	Missense mutations and deletions [91].	Alpha-actinin-4 [91]	Adult-onset familial nephrotic syndrome; typically resistant to corticosteroids; associated with FSGS [91].	Actin stabilization therapies, early monitoring, and personalized treatment plans [91].
*TRPC6*	Gain-of-function mutations and missense mutations [46,91].	TRPC6 [46,91]	Familial FSGS; proteinuria due to increased calcium influx; progressive renal disease [46,91].	TRPC6 inhibitors, targeted therapies to reduce podocyte calcium levels, and gene editing [46,91].
*CD2AP*	Deletions and loss-of-function mutations [42,90,91].	CD2-associated protein [42,90,91]	Steroid-resistant nephrotic syndrome, podocyte injury, and risk of progressing to ESRD [42,90,91].	CD2AP stabilization strategies, gene therapy, and personalized immunomodulation [34,90,91].
*NEPH1*	Missense mutations and nonsense mutations [34].	Nephrin-like protein 1 [34]	Like NPHS1, involved in slit diaphragm function; associated with severe proteinuria [34].	Gene therapy, NEPH1 stabilization, and nephrin pathway-targeted therapies [34].
*PLCE1*	Homozygous mutations and deletions [92].	Phospholipase C epsilon [92]	Early-onset nephrotic syndrome, rapid progression to ESRD, and variable response to steroids [92].	PLCE1-targeted therapies, early genetic screening, and personalized care strategies [92].

End-stage renal disease [ESRD]; Focal segmental glomerulosclerosis [FSGS].

**Table 6 ijms-25-12174-t006:** Therapeutic response based on genetic and biomarker profiling in MCD.

Genetic Mutation/Biomarker	Associated Treatment Response	Therapeutic Recommendations	Clinical Implications
NPHS1 Mutation	Poor response to corticosteroids and higher risk of steroid resistance [42,90,91].	Early use of alternative immunosuppressants; gene therapy (in development) [42,90,91].	Indicates need for genetic screening; may benefit from personalized therapy [42,90,91].
NPHS2 Mutation	Increased risk of progression to end-stage renal disease and poor response to standard therapies [42,90,91].	Consider early aggressive treatment; potential for gene therapy [42,90,91].	Suggests close monitoring and early intervention [42,90,91].
WT1 Mutation	Association with steroid resistance and risk of Wilms tumor [91].	Early genetic counseling, tumor surveillance, and alternative therapies [91].	High-risk group requiring specialized management [91].
Urinary Nephrin Levels	Prominent levels indicate active podocyte injury, correlating with disease severity.	Use as a biomarker for treatment adjustment, particularly in relapse cases.	Helps in monitoring treatment efficacy and disease activity.
Podocalyxin Levels	Elevated levels suggest ongoing podocyte detachment and damage [62,98].	Consider adjusting therapy to target podocyte health [62,98].	Biomarker for active disease, useful in predicting relapse [62,98].
Urinary CD80 Levels	Elevated levels may predict poor response to steroids [51,63].	Early consideration of rituximab or other B-cell-depleting therapies [51,63].	Potential for personalized immunomodulatory treatment [63].
Alpha-1-microglobulin Levels	Elevated levels associated with tubular damage and poor long-term outcomes [59,99].	Focus on nephroprotective strategies; potential for novel tubular-targeted therapies [59,99].	Indicators of overall kidney health help in long-term prognosis [59,99].

**Table 7 ijms-25-12174-t007:** Comparison of immune dysregulation in MCD and other nephrotic syndromes.

Disease.	Key Immune Dysregulation Features	Cytokine Profiles	Typical Immune Responses	Therapeutic Implications
Minimal Change Disease	T-cell abnormalities and autoantibodies against nephrin [47,60].	Elevated IL-13, TNF-alpha, and IL-4 levels [47,60].	Th2-mediated response; podocyte-targeted damage [47,60].	Corticosteroids, rituximab, and anti-IL-13 therapies [47,60].
Focal Segmental Glomerulosclerosis	Circulating permeability factors and possible genetic predispositions [104].	Elevated suPAR, TNF-alpha, and IL-6 levels [104].	Immune-mediated podocyte damage; often steroid-resistant [104].	Calcineurin inhibitors, rituximab, and plasmapheresis [104].
Membranous Nephropathy	Autoantibodies against PLA2R and immune complex deposition [60].	Elevated IL-6, TNF-alpha, and IL-1 levels [60].	Immune complex-mediated glomerular injury; complement activation [60].	Rituximab, cyclophosphamide, and anti-complement therapies [60].
IgA Nephropathy	Aberrant IgA production and immune complex deposition in mesangium [104,105].	Elevated IL-6 and TGF-beta levels [104,105].	Complement-mediated mesangial injury; immune complex deposition [104,105].	Corticosteroids, ACE inhibitors, and mycophenolate mofetil [104,105].
Lupus Nephritis	Autoantibodies against dsDNA and immune complex deposition in glomeruli [106].	Elevated IL-6, IL-10, TNF-alpha, and interferon-gamma levels [106].	Immune complex-mediated glomerular and tubulointerstitial damage [106].	Corticosteroids, mycophenolate mofetil, biologics targeting TNF-alpha, and IL-6 [106].

Soluble urokinase plasminogen activator receptor [suPAR].

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
