# Peer review of "Detailed Pathophysiology of Minimal Change Disease: Insights into Podocyte Dysfunction, Immune Dysregulation, and Genetic Susceptibility"

_ijms, 2024, doi:10.3390/ijms252212174_

Round 1
Reviewer 1 Report
Comments and Suggestions for Authors
To thank the authors for the excellent work as a narrative review of the detailed pathophysiology of MCD, emphasizing the dysfunction, inmune dysregulation and genetic susceptibility.
A clear, concise, separate manuscript on important aspects of the pathophysiology of this disease.
Just give these suggestions for improving the manuscript:
-The text from line 127 to 129 is repeated in several paragraphs, please modify them.
-Line 129 to 131: What types of therapies are they ? add a comment on it
-Line 167 to 169: What types of immunomodulatory therapies are ? add a comment about it.
-In figure 1, the legend is missing, add it.
For the rest of the manuscript I have no further suggestions.
Congratulations for the work
Reviewer 2 Report
Comments and Suggestions for Authors
1. To improve the educational value of the paper, the incidence and prevalence of MCD should be presented.
2. Some content is repeated: lines 102-118 and lines 282-289. Please, correct.
3. The paragraph on the biomarkers should be moved before the paragraph on the treatment.
4. Line 153: Please, specify which study do you mean?
5. Line 202: Please, explain in more detail what is the connection between MCD and FSGS.
6. Line 219: avoid brand names of medicines (Myfortic).
7. The references should be cited in order of their appearance. Please, correct.
8. Figure 1 should be cited in the text.
9. Figure 1 is lacking the legend.
10. Tables should be cited in the text.
11. Some spaces are lacking, e.g.: line 39 (inchildren), line 40 (inadults), line 41 (suddenonset), line 44 (thatare), line 461 (geneenvironement)
